# Evidence of leaky protection following COVID-19 vaccination and SARS-CoV-2 infection in an incarcerated population

Margaret L. Lind [1] ✉, Murilo Dorion[1], Amy J. Houde[2], Mary Lansing[2], Sarah Lapidus[1], Russell Thomas[1], Inci Yildirim [1,3], Saad B. Omer[1,4,5], Wade L. Schulz [6,7], Jason R. Andrews [8], Matt D. T. Hitchings [9], Byron S. Kennedy[2], Robert P. Richeson[2], Derek A. T. Cummings [10,11,13] & Albert I. Ko [1,12,13] ✉

Whether SARS-CoV-2 infection and COVID-19 vaccines confer exposure-dependent ("leaky") protection against infection remains unknown. We examined the effect of prior infection, vaccination, and hybrid immunity on infection risk among residents of Connecticut correctional facilities during periods of predominant Omicron and Delta transmission. Residents with cell, cellblock, and no documented exposure to SARS-CoV-2 infected residents were matched by facility and date. During the Omicron period, prior infection, vaccination, and hybrid immunity reduced the infection risk of residents without a documented exposure (HR: 0.36 [0.25–0.54]; 0.57 [0.42–0.78]; 0.24 [0.15–0.39]; respectively) and with cellblock exposures (0.61 [0.49–0.75]; 0.69 [0.58–0.83]; 0.41 [0.31–0.55]; respectively) but not with cell exposures (0.89 [0.58–1.35]; 0.96 [0.64–1.46]; 0.80 [0.46–1.39]; respectively). Associations were similar during the Delta period and when analyses were restricted to tested residents. Although associations may not have been thoroughly adjusted due to dataset limitations, the findings suggest that prior infection and vaccination may be leaky, highlighting the potential benefits of pairing vaccination with non-pharmaceutical interventions in crowded settings.

A fundamental question regarding SARS-CoV-2 immunity is whether infection and vaccination confer all-or-nothing or exposure-dependent ("leaky") protection against infection. Despite continued evidence that prior SARS-CoV-2 infections and COVID-19 vaccines provide protection against infection and COVID-19 related illness, protection is incomplete[1–8]. While key reasons for imperfect protection include waning protection and variant-specific immune evasion, differences in the viral dose during an infectious exposure may also contribute[6,9–16]. In alignment with this hypothesis, the immunity conferred by prior SARS-CoV-2 infection and COVID-19 vaccination have

[1]Department of Epidemiology of Microbial Diseases, Yale School of Public Health, New Haven, CT, USA. [2]Connecticut Department of Correction, Wethersfield, CT, USA. [3]Department of Pediatrics, Yale School of Medicine, New Haven, CT, USA. [4]Yale Institute for Global Health, Yale School of Public Health, New Haven, CT, USA. [5]UT Southwestern, School of Public Health, Dallas, TX, USA. [6]Department of Internal Medicine, Yale School of Medicine, New Haven, CT, USA. [7]Department of Laboratory Medicine, Yale University School of Medicine, New Haven, CT, USA. [8]Division of Infectious Diseases and Geographic Medicine, Stanford University, Stanford, CA, USA. [9]Department of Biostatistics, College of Public Health & Health Professions, University of Florida, Gainesville, FL, USA. [10]Department of Biology, University of Florida, Gainesville, FL, USA. [11]Emerging Pathogens Institute, University of Florida, Gainesville, FL, USA. [12]Instituto Gonçalo Moniz, Fundação Oswaldo Cruz, Salvador, BA, Brazil. [13]These authors contributed equally: Derek A. T. Cummings, Albert I. Ko. ✉e-mail: Margaret.Lind@yale.edu; Albert.Ko@yale.edu

been speculated to be "leaky", whereby protection reduces infection risk on a per-exposure basis[17–20]. While there are examples of leaky vaccines for infectious diseases, including the RTS,S/ASO1 vaccine for malaria[20,21] and attenuated vaccines for Marek's disease[22], empirical evidence for this phenomenon has not been reported for prior SARS-CoV-2 infection and COVID-19 vaccines.

The key barrier to interrogating leaky protection in SARS-CoV-2 immunity is the inherent challenge of measuring viral dose, whether incident or cumulative over time. Investigations thus rely on evaluating proxies such as proximity and duration of exposure to an infected index case. Yet, the use of such proxies has been limited by the lack of reliable information at required scales and by misclassification due to movement and social interactions in real world settings.

The controlled social structure of correctional facilities provides an opportunity to address these limitations and delineate whether prior SARS-CoV-2 infection and COVID-19 vaccination confer leaky protection. As a result of the defined housing of residents, residents can be classified as having close exposures (within cell), moderate exposures (within cellblock), or no documented exposures to a SARS-CoV-2 infected resident on a given day. These exposure categories can serve as a proxy for exposure risk in a high transmission setting where movement is restricted between spatial units. Herein, we leveraged the ability to classify residents by recent SARS-CoV-2 exposures and the high frequency of testing performed by the Connecticut Department of Correction (DOC) to compare the risk of infection and effects of prior infection, vaccination, and hybrid immunity (prior infection and vaccination) among residents with cell, cellblock, and no documented exposures to SARS-CoV-2 infected residents during the periods of Delta and Omicron predominance in Connecticut, USA.

## Results

### COVID-19 prevention and SARS-CoV-2 infections in the correctional system

The Connecticut DOC system is comprised of 13 facilities with a daily census of ~9300 residents[23]. During the study (June 15, 2021 and May, 10, 2022), a total of 15,444 people spent at least one night housed in a DOC-operated facility, of which, 13,490 and 11,492 were residents during periods, respectively, of predominant Delta variant (June 15 to December 12, 2021) and Omicron variant (December 13, 2021 to May 10, 2022) transmission in Connecticut[24]. As of the end of the study, 48% of currently incarcerated residents had completed their primary vaccine series and 27% had received a booster dose (Fig. 1A).

The DOC implemented a SARS-CoV-2 testing program consisting of testing of residents who were symptomatic, were contacts of confirmed cases, were due in court or had employment required testing, and residents who were newly incarcerated or transferred between facilities (rapid antigen testing). In addition, the DOC conducted voluntary, bi-weekly mass screening of 10% of residents (RT-PCR testing). Contact tracing included testing residents of (1) the same cell as an infected resident or (2) the same cellblock or facility as an infected resident if close contact (being within six feet for ≥15 min within a 24-h period) was reported by the infected resident (see Supplement DOC COVID-19 Testing). In total, 87,884 SARS-CoV-2 tests were performed during the study period, of which 20,794 were RT-PCRs and 67,090 were rapid antigen tests (Fig. 1B). Contact tracing among residents without reported symptoms comprised the largest proportion of testing (54%) followed by mass screening (24%; Fig. 1B). On average, the DOC tested 25% of residents every 2 weeks and 65% every 3 months during the study period.

Testing intensified from November 2021 to February 2022 (Fig. 1B) when Delta and Omicron BA.1 variant transmission contributed to an epidemic wave in Connecticut. During this period, the average proportion of residents tested in a 14-day period was 33.6% (red line, Fig. 1C). A total of 5079 SARS-CoV-2 infections were identified, of which 1598 and 3481 occurred during the Delta and Omicron periods, respectively. Among the 5079 infections, 57% and 38% were identified through contact tracing among residents without reported symptoms and testing in the presence of recorded symptoms, respectively (Fig. 1D).

### Rolling matched cohort of residents exposed to SARS-CoV-2 infection

We conducted a rolling matched cohort study that compared the risk of SARS-CoV-2 infection and effectiveness of prior infection, vaccination, and hybrid immunity among residents with cell, cellblock, and without documented exposures to an infected case (Supplement Fig. 1). A cell exposure event was defined as having ≥1 cellmate test positive for SARS-CoV-2 in the absence of cellmates testing positive in the prior 14 days. A cellblock exposure event was defined as having ≥1 resident of the same cellblock (but different cell) test positive in the absence of a cellmate or resident of the cellblock testing positive in the prior 14 days. Events without documented exposures were defined as days when residents did not have a cell or cellblock exposure event in the prior 14 days. We prevented the inclusion of multiple events without documented exposures from the same person during a 14-day period through random selection. We selected a cohort of events by cluster matching on facility and calendar day and ascertained infection in the subsequent 14-day period during Delta and Omicron periods.

During the Delta period, we identified 290 cell and 5805 cellblock exposure events among the 7389 residents who were incarcerated for ≥14 days and spent ≥1 night in a cell with a roommate (Fig. 2A). Among the 584,629 events without documented exposures, we randomly selected 37,394 unique events. After matching, we identified a sample of 264 cell exposure events (258 residents), 5,616 cellblock exposure events (3745 residents), and 17,024 events without documented exposure (6073 residents).

During the Omicron period, we identified 796 cell and 6408 cellblock exposure events and 259,320 events without documented exposures among 6161 residents who were incarcerated for ≥14 days and resided in a cell with a roommate for ≥1 day (Fig. 2B). We randomly selected 20,125 of 259,320 events without a documented exposure. Following matching, we selected 702 cell exposure events (671 residents), 5980 cellblock exposure events (4135 residents), and 13,464 events without documented exposures (5429 residents).

### Characteristics of residents with and without exposure to SARS-CoV-2 infections

During the Delta period, events with and without documented exposures occurred among racially similar residents and residents with similar cell sizes (median: 2 residents). However, cellblock exposure events occurred more frequently among residents of larger median cellblock sizes (107.0 residents) than cell exposure events (74 residents) or events without documented exposures (88 residents). Cell exposure events occurred less frequently among people with recorded prior infections (32.2%), vaccination (41.3%), or hybrid immunity (17.1%) than events without documented exposures (infection, 38.8%; vaccination, 53.7%; hybrid, 25.2%; Table 1). Male residents were more likely to have had a prior, recorded SARS-CoV-2 infection than female residents regardless of their SARS-CoV-2 exposure status (Supplement Table 1). Among residents of the same age, race, room-size, cellblock, and inclusion time, the time since last prior infection and vaccination did not differ significantly between residents with and without documented exposures (Supplement Table 2).

During the Omicron period, events with and without documented exposures occurred among racially similar residents and residents with similar cell and cellblock sizes. Cell exposure events occurred with similar frequency among unvaccinated residents (46.0%) as cellblock exposure events (43.1%) and events without documented exposures (42.9%). Cell exposure events occurred less frequently among people with recorded prior infections (36.2%) or

hybrid immunity (23.9%) than cellblock exposure events (prior infection: 43.6%; hybrid immunity: 28.9%) or events without documented exposures (prior infection: 47.0%; hybrid immunity: 30.4%; Table 1). Male residents were more likely to have had a prior, recorded SARS-CoV-2 infection than female residents regardless of their exposure status (Supplement Table 1). Among residents of the same age, race, room-size, cellblock, and inclusion time, the time since last prior infection and vaccination did not differ significantly between residents with and without documented exposures (Supplement Table 2).

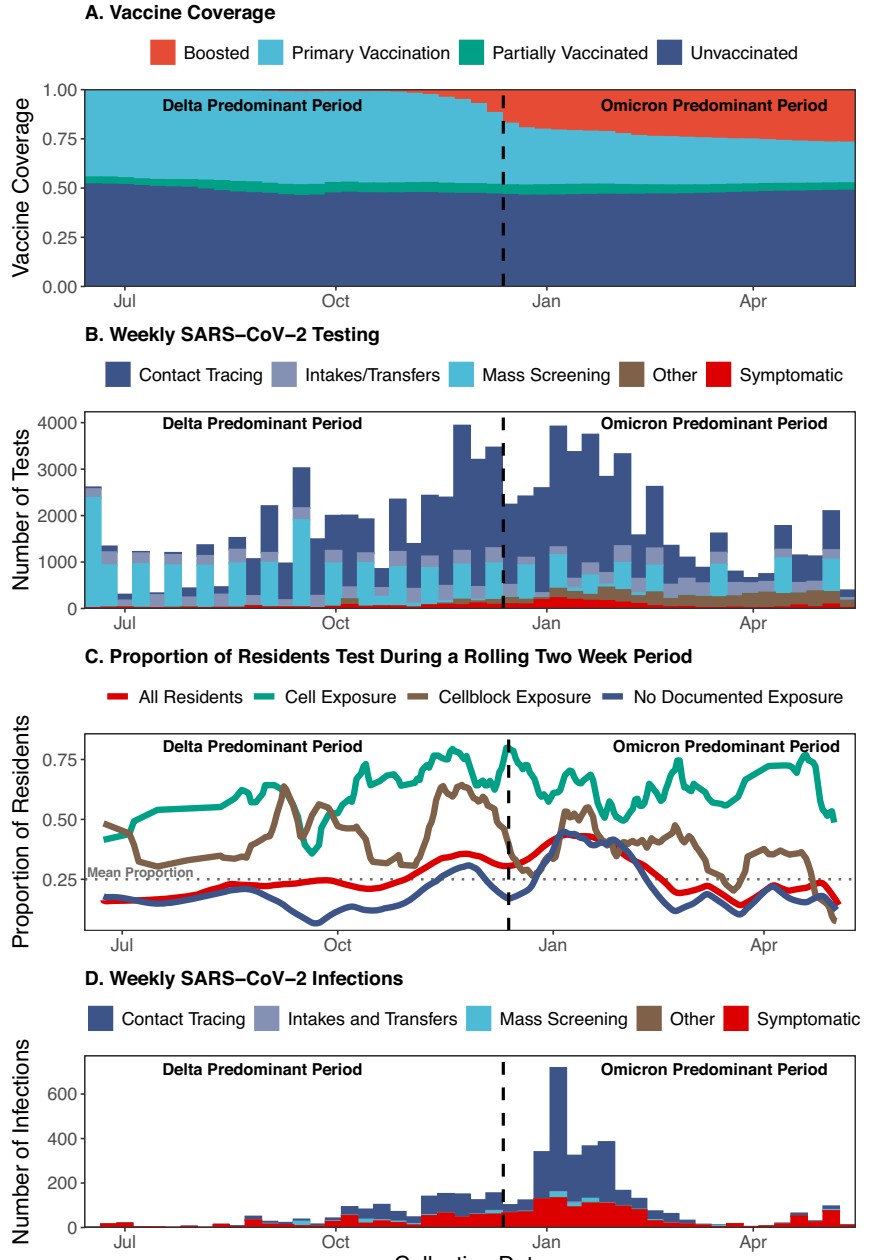

**Fig. 1 | Vaccination coverage, SARS-CoV-2 testing, proportion of res0idents Tested, and SARS-CoV-2 infections in the Connecticut Correctional Facility system between June 15, 2021, and May 10, 2022.** The (**A**) vaccination coverage (red: boosted, light blue: primary vaccination, green: partially vaccinated, navy: unvaccinated), (**B**) number of SARS-CoV-2 tests conducted as part of mass screening (light blue), contact tracing in the absence of recorded symptoms (navy), intake/transfer testing in the absence of recorded symptoms (grey/blue), other testing in the absence of recorded symptoms (brown), and testing in the presence of recorded symptoms (symptoms data not available for mass screening [PCR] testing; red), (**C**) proportion of residents tested during a rolling 14-day period among all residents (red) and residents with cell exposure events (green), cellblock exposure events (brown), and no documented exposure events (navy), (**D**) number of SARS-CoV-2 infections detected as part of mass screening (light blue), contact tracing in the absence of recorded symptoms (navy), intake/transfer testing in the absence of recorded symptoms (grey/blue), other testing in the absence of recorded symptoms (brown), and testing in the presence of recorded symptoms (red) among people who resided in Connecticut Department of Correction Facility cells between June 15, 2021, and May 10, 2022. During the study period, RT-PCR tests were collected for mass testing and rapid antigen tests were collected for the following primary reasons: intakes/transfer, contact tracing, symptom presence, and employment. Infections were defined as a positive test (RT-PCR or rapid antigen test) collected in the absence of a positive test in the last 90 days. Residents were classified as having a cell exposure event on the day their cellmate tested positive, having a cellblock exposure event the day a resident of their cellblock but not cell tested positive, and having an event without documented exposures if no one in their cellblock tested positive on a given day.

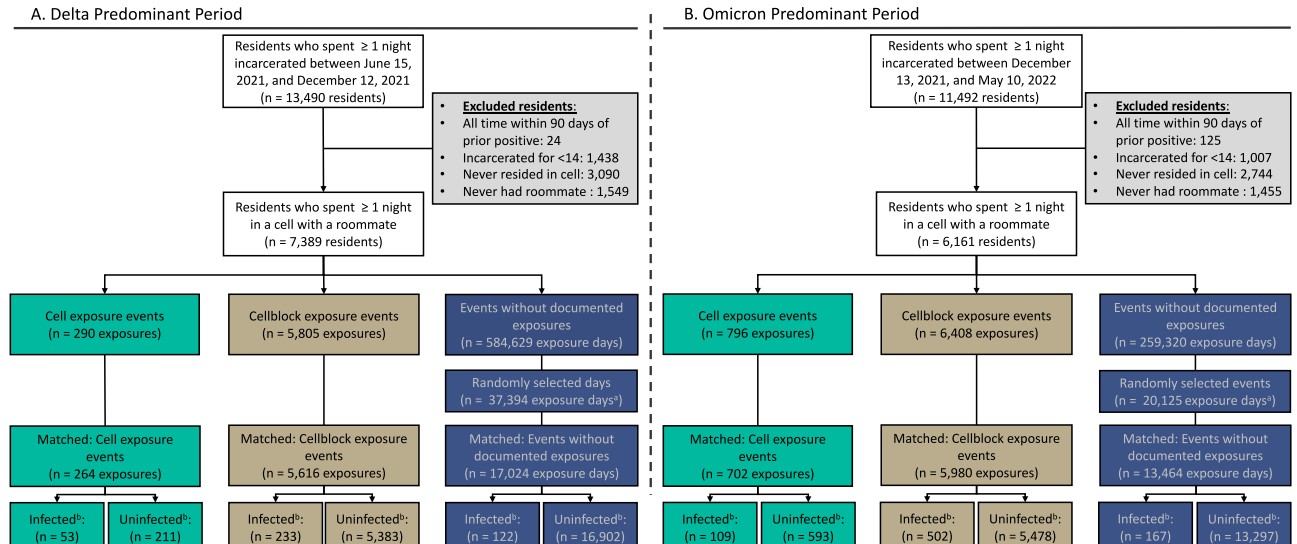

**Fig. 2 | Selection of the rolling matched cohort of residents according to their facility exposures.** Flowchart showing how people incarcerated within Connecticut Department of Correction facilities and who resided in cells between June 15, 2021, and December 12, 2021 (Delta Predominant Period [A]) and December 13, 2021, and May 10, 2022 (Omicron Predominant Period [B]), were included in the analysis. Residents were classified as having a cell exposure event (green) on the day their cellmate tested positive, having a cellblock exposure event (brown) the day a resident of their cellblock but not cell tested positive, and having an event without documented exposure (navy) if no one in their cellblock tested positive on a given day. Cell exposure events that occurred within 14 days following a prior cell exposure event were excluded. Cellblock exposure events and events without documented exposures that occurred in the 14 days following a cellblock or cell exposure event were excluded. **A** To prevent the inclusion of multiple events without documented exposures from the same person during a 14-day period, we randomly selected incarceration events without documented exposures and excluded all others within the prior or following 14 days. **B** We defined infections as a positive RT-PCR or rapid antigen test during the 14 days of follow-up.

## High exposure settings in cells and cellblocks impart increased infection risk

During the Delta period, 122 residents tested positive following an event without documented exposure, 233 residents tested positive following a cellblock exposure event, and 53 residents tested positive following a cell exposure event (Fig. 2A). The hazard of infection was 2.67 (95% Confidence Interval [CI]: 1.84–3.88) and 9.70 (6.29–14.96) times higher following cellblock and cell exposure events than events without documented exposure, respectively (Fig. 3). The hazard of symptomatic infection, defined as a positive rapid antigen test collected from a symptomatic resident, was 2.21 (1.28–3.82) and 7.44 (3.87–14.30) times higher following cellblock and cell exposure events than events without documented exposure, respectively (Supplement Fig. 2; Unadjusted estimates: Supplement Table 4).

We conducted a series of sensitivity analyses to address concerns regarding potential sources of bias (Supplement: Sensitivity Analyses). Of primary concern is the bias resulting from inequal testing following events with and without documented exposure (Fig. 1C; Supplement Fig. 4). To examine these biases, we performed sensitivity analyses (1) restricted to residents who were tested during follow-up and (2) restricted to residents tested during follow-up for non-symptomatic reasons. Following the restriction to tested residents, the hazard of infection was 1.89 (1.36–2.64) and 5.23 (3.50–7.82) times higher following cellblock and cell exposure events than events without documented exposures, respectively (Supplement Fig. 5; Unadjusted estimates: Supplement Table 9). Restricting to tests conducted for non-symptomatic reasons resulted in the exclusion of an additional 12 facility exposure events and the point estimates were within 0.01 of the sensitivity analysis restricting to tested residents (Supplement Fig. 7; Unadjusted estimates: Supplement Table 11).

Additionally, we did not have access to community infection data and were concerned about the bias introduced from prior infection misclassification. To reduce this bias, we conducted a sensitivity analysis limited to people incarcerated since the beginning of our study (June 15, 2021). The hazard of infection was 3.15 (2.01–4.92) and 12.96 (7.90–21.26) times higher following cellblock and cell exposure events than events without documented exposure, respectively (Supplement Fig. 9; Unadjusted estimates: Supplement Table 13). Further, since we may have overestimated the effect of facility exposures by including residents who were exposed to more than one index case on a given day, we conducted a sensitivity analysis limited to cellblock and cell exposure events with only one index case. To ensure including already infected residents did not drive our findings, we conducted an analysis restricted to residents who tested negative in the prior 5 days. To ensure our exposures were temporally linked to observed infections, we conducted two sensitivity analyses: one excluding the first 2 days of follow-up, and one limiting follow-up to 9 days. We found that cellblock and cell exposure events were significantly associated with the hazards of infection for each scenario (Supplement Figs. 11, 13, 14, 16; Unadjusted estimates: Supplement Tables 15, 17, 18, 20).

During the Omicron period, 167 residents tested positive following an event without documented exposures, 502 residents tested positive following a cellblock exposure event, and 109 residents tested positive following a cell exposure event (Fig. 2B). The hazard of infection was 3.34 (2.22–5.00) and 4.73 (3.05–7.36) times higher following cellblock or cell exposure events than events without documented exposure, respectively (Fig. 3). The hazard of symptomatic infection was 3.82 (2.08–7.00) and 7.00 (3.61–13.58) times higher following cellblock and cell exposure events than events without documented exposure, respectively (Supplement Fig. 2; Unadjusted estimates: Supplement Table 4).

We conducted the same sensitivity analyses as for the Delta period. Following the restriction to tested residents, the hazard of infection was 2.14 (1.62–2.82) and 2.23 (1.62–3.07) times higher following cellblock and cell exposure events than events without documented exposure, respectively (Supplement Fig. 5; Unadjusted estimates: Supplement Table 9). Following the restriction to residents incarcerated since the beginning of the study, the hazard of infection was 4.40 (2.84–6.82) and 6.17 (3.75–10.14) times higher following cellblock and cell exposure events than events without documented exposure

(Supplement Fig. 9; Unadjusted estimates: Supplement Table 13). Cellblock and cell exposures were found to be significantly associated with an increased hazard of infection for each additional scenario (Supplement Figs. 7, 11, 13, 14, 16; Unadjusted estimates: Supplement Tables 11, 15, 17, 18, 20).

## High exposure setting overcomes the protection afforded by infection, vaccination, and hybrid immunity

During the Delta period, the effectiveness of prior infection at reducing the hazard of SARS-CoV-2 infection was highest following events without documented exposure (Hazard Ratio [HR]: 0.21 [0.11–0.39]) and lowest following cell exposure events (HR: 0.59 [0.30–1.16]). Vaccine effectiveness was highest following events without documented exposure (HR: 0.32 [0.21–0.49]) and lowest following cell exposure events (HR: 0.74 [0.37–1.48]). The effectiveness of hybrid immunity was highest following events without documented exposure (HR: 0.05 [0.02–0.10]) and lowest following cell exposure events (HR: 0.29 [0.07–1.12]). The effectiveness of prior infection, vaccination, and hybrid immunity was significantly lower following cell exposure events than following events without documented exposure (P = 0.029, 0.033, 0.026, respectively; Fig. 4/Supplement Table 6). The effectiveness of prior infection and vaccination at reducing the hazard of symptomatic SARS-CoV-2 infection was highest following events without documented exposure (HR: infection, 0.18 [0.07–0.45], vaccination, 0.21 [0.11–0.41]) and lowest following cell exposure events (HR: infection, 0.42 [0.11–1.60], vaccination, 0.53 [0.17–1.64]). No residents with hybrid immunity had a symptomatic infection following a cell exposure event (Supplement Fig. 3; Unadjusted estimates: Supplement Table 5).

We performed sensitivity analyses that paralleled those described above (see Supplement: Sensitivity Analyses). The effectiveness of prior infection and hybrid immunity was highest following events without documented exposure and lowest following cell exposure events under all scenarios (Supplement Figs. 6, 8, 10, 12, 15, 17; Unadjusted estimates: Supplement Tables 10, 12, 14, 16, 19, 21). Vaccine effectiveness was highest following events without documented exposures for all scenarios, except when we limited follow-up to 9 days (Supplement Fig. 17; Unadjusted estimates: Supplement Table 21). Under this scenario, vaccination reduced the hazard of infection by 0.31 (0.19–0.51) times following events without documented exposure, 0.31 (0.21–0.46) times following cellblock exposure events, and 0.87 (0.45–1.69) times following cell exposure events. When we restricted to residents tested during follow-up, the effectiveness of prior infection, vaccination, and hybrid immunity was highest following events without documented exposure (HR: infection, 0.23 [0.12–0.42]; vaccination, 0.34 [0.22–0.52]; hybrid, 0.05 [0.02–0.11]) and lowest following cell exposure events (HR: infection, 0.50 [0.25–0.98]; vaccination, 0.72 [0.37–1.41]; hybrid, 0.33 [0.12–0.91]; Supplement Fig. 6; Unadjusted estimates: Supplement Table 10). Following the restriction to people incarcerated since the beginning of the study, the effectiveness of prior infection, vaccination, and hybrid immunity was highest following events without documented exposure (HR: infection, 0.25 [0.13–0.48]; vaccination, 0.32 [0.19–0.54]; hybrid, 0.07 [0.02–0.18]) and lowest following cell exposure events (HR: infection, 0.51 [0.25–1.03]; vaccination, 0.77 [0.38–1.58]; hybrid, 0.31 [0.10–0.99]; Supplement Fig. 10; unadjusted estimates: Supplement Table 14).

During the Omicron period, the effectiveness of prior infection was highest following events without documented exposure (HR: 0.36 [0.25–0.54]) and lowest following cell exposure events (HR: 0.89 [0.58–1.35]). Vaccine effectiveness was highest following events without documented exposure (HR: 0.57 [0.42–0.78]) and lowest following cell exposure events (HR: 0.96 [0.64–1.46]). The effectiveness of hybrid immunity was highest following events without documented exposure (HR: 0.24 [0.15–0.39]) and lowest following cell exposure

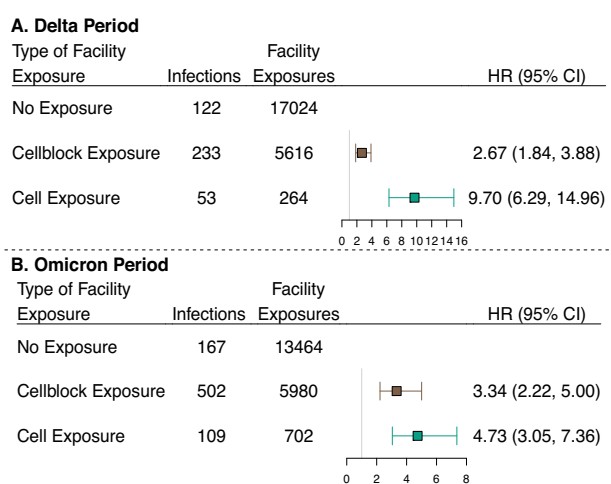

**A. Delta Period**

| Type of Facility Exposure | Infections | Facility Exposures | HR (95% CI) |
|---|---|---|---|
| No Exposure | 122 | 17024 | |
| Cellblock Exposure | 233 | 5616 | 2.67 (1.84, 3.88) |
| Cell Exposure | 53 | 264 | 9.70 (6.29, 14.96) |

**B. Omicron Period**

| Type of Facility Exposure | Infections | Facility Exposures | HR (95% CI) |
|---|---|---|---|
| No Exposure | 167 | 13464 | |
| Cellblock Exposure | 502 | 5980 | 3.34 (2.22, 5.00) |
| Cell Exposure | 109 | 702 | 4.73 (3.05, 7.36) |

**Fig. 3 | Association between documented exposure and SARS-CoV-2 infection risk among residents of Connecticut Department of Correction facilities between June 15, 2021, and May 10, 2022.** Forest plot depicting the association between documented close exposure to a SARS-CoV-2 infected resident and the risk of subsequent SARS-CoV-2 infection. Residents were classified as having a cell exposure event (green) on the day their cellmate tested positive, having a cellblock exposure event (brown) the day a resident of their cellblock but not cell tested positive, and having an event without documented exposure if no one in their cellblock tested positive on a given day. Cell exposure events that occurred within 14 days following a prior cell exposure event were excluded. Cellblock exposure events and events without documented exposures that occurred in the 14 days following a cellblock or cell exposure event were excluded. Facility exposures were stratified by periods of variant predominance (Delta [**A**]: June 15, 2021–December 12, 2021; Omicron [**B**]: December 13, 2021–May 10, 2022). The associations were estimated using a Cox Proportional Hazard Model stratified by facility and with robust standard errors. The model was adjusted for age, calendar date, race, room and cellblock size, vaccination, and prior infection status of the susceptible person. Boxes indicate estimated hazard ratio (HR) point values and whiskers indicate 95% confidence intervals (Delta Period: n = 22,904 facility events; Omicron Period: n = 20,146 facility events). Unadjusted results presented in Supplement Table 3.

events (HR: 0.80 [0.46–1.39]). The effectiveness of prior infection, vaccination, and hybrid immunity was significantly lower following cell exposure events than events without documented exposure (P = 0.002, 0.041, and 0.001, respectively; Fig. 4/Supplement Table 6). The effectiveness of prior infection, vaccination, and hybrid immunity at reducing the hazard of symptomatic SARS-CoV-2 infection was highest following events without documented exposure (HR: infection, 0.35 [0.21–0.59]; vaccination, 0.33 [0.21–0.53]; hybrid, 0.13 [0.06–0.28]) and lowest following cell exposure events (HR: infection, 0.77 [0.45–1.31]; vaccination, 0.62 [0.35–1.10]; hybrid, 0.53 [0.26–1.11]; Supplement Fig. 3; Unadjusted estimates: Supplement Table 5).

Following the restriction to residents tested during follow-up, the effectiveness of prior infection, vaccination, and hybrid immunity was highest following events without documented exposures (HR: infection, 0.44 [0.30–0.63]; vaccination, 0.49 [0.36–0.68]; hybrid 0.32 [0.19–0.53]) and lowest following cell exposure events (HR: infection, 0.69 [0.47–1.02]; vaccination, 0.81 [0.53–1.22]; hybrid, 0.67 [0.44–1.02]; Supplement Fig. 6; unadjusted estimates: Supplement Table 10). Following the restriction to people incarcerated since the study began, the effectiveness of prior infection, vaccination, and hybrid immunity was highest following events without documented exposures (HR: infection, 0.38 [0.24–0.58]; vaccination, 0.61 [0.43–0.86]; hybrid, 0.23 [0.14–0.38]) and lowest following cell exposure events (HR: infection, 0.88 [0.56–1.37]; vaccination, 0.79 [0.49–1.28]; hybrid, 0.64 [0.39–1.05]; Supplement Fig. 10; Unadjusted estimates: Supplement Table 14). The effectiveness of prior infection, vaccination, and hybrid immunity was highest following events

**Table 1 | Characteristics of residents with and without documented exposure to SARS-CoV-2 infected residents**

| Characteristics | Delta predominant period (June 15, 2021–December 12, 2022) | | | Omicron predominant period (December 13, 2021–May 10, 2022) | | |
|---|---|---|---|---|---|---|
| | Cell exposure events (N = 264) | Cellblock exposure events (N = 5616) | Events without documented exposures (N = 17024) | Cell exposure events (N = 702) | Cellblock exposure events (N = 5980) | Events without documented exposures (N = 13464) |
| Age (Median [Qr 1-3]) | 34 [27, 43] | 36 [29, 46] | 36 [29, 46] | 37 [30, 46] | 36 [28, 45] | 36 [29, 46] |
| Gender Female (N [%]) | 38 (14.4%) | 383 (6.8%) | 1475 (8.7%) | 111 (15.8%) | 874 (14.6%) | 1463 (10.9%) |
| Race & Ethnicity (N [%]) | | | | | | |
| Non-Hispanic Black | 126 (47.7%) | 2484 (44.2%) | 7736 (45.4%) | 318 (45.3%) | 2644 (44.2%) | 6052 (44.9%) |
| Non-Hispanic White | 69 (26.1%) | 1482 (26.4%) | 4354 (25.6%) | 210 (29.9%) | 1753 (29.3%) | 3987 (29.6%) |
| Other (Hispanic, American Indian, & Asian) | 69 (26.1%) | 1650 (29.4%) | 4934 (29.0%) | 174 (24.8%) | 1583 (26.5%) | 3425 (25.4%) |
| Duration of Incarceration (Median [QR 1-3])[a] | 181 [154, 181] | 181 [181, 181] | 181 [181, 181] | 149 [149, 149] | 149 [149, 149] | 149 [149, 149] |
| Cell Size (Median [QR 1-3])[b] | 2 [2, 2] | 2 [2, 2] | 2 [2, 2] | 2 [2, 2] | 2 [2, 2] | 2 [2, 2] |
| Cellblock Size (Median [QR 1-3])[b] | 74 [56, 129] | 107 [69, 155] | 88 [65, 140] | 82 [61, 104] | 89 [67, 133] | 89 [65, 120] |
| History of Prior Infection (N [%])[c] | 85 (32.2%) | 2180 (38.8%) | 6522 (38.3%) | 254 (36.2%) | 2606 (43.6%) | 6329 (47.0%) |
| History of Vaccination (N [%])[c] | 109 (41.3%) | 3013 (53.7%) | 9141 (53.7%) | 379 (54.0%) | 3401 (56.9%) | 7693 (57.1%) |
| History of Hybrid Immunity (N [%])[c] | 45 (17.0%) | 1379 (24.6%) | 4289 (25.2%) | 168 (23.9%) | 1729 (28.9%) | 4095 (30.4%) |
| Follow-up Time (Median [Qr 1-3]) | 14 [14, 14] | 14 [14, 14] | 14 [4, 14] | 14 [14, 14] | 14 [14, 14] | 9 [2, 14] |

[a] Duration of incarceration (in days) including the SARS-CoV-2 exposure event type.

[b] Number of cellmates or unit-mates at the time of the exposure event that resulted in study inclusion.

[c] History of prior infection defined as a record of prior infection (defined as prior infection defined as the record of a positive SARS-CoV-2 test (rapid antigen or RT-PCR) at least 90 days prior to follow-up start date); history of vaccination defined as receipt of at least one vaccine dose as of the start of follow-up: hybrid immunity defined as a recorded prior infection and vaccination as of the start of the study period.

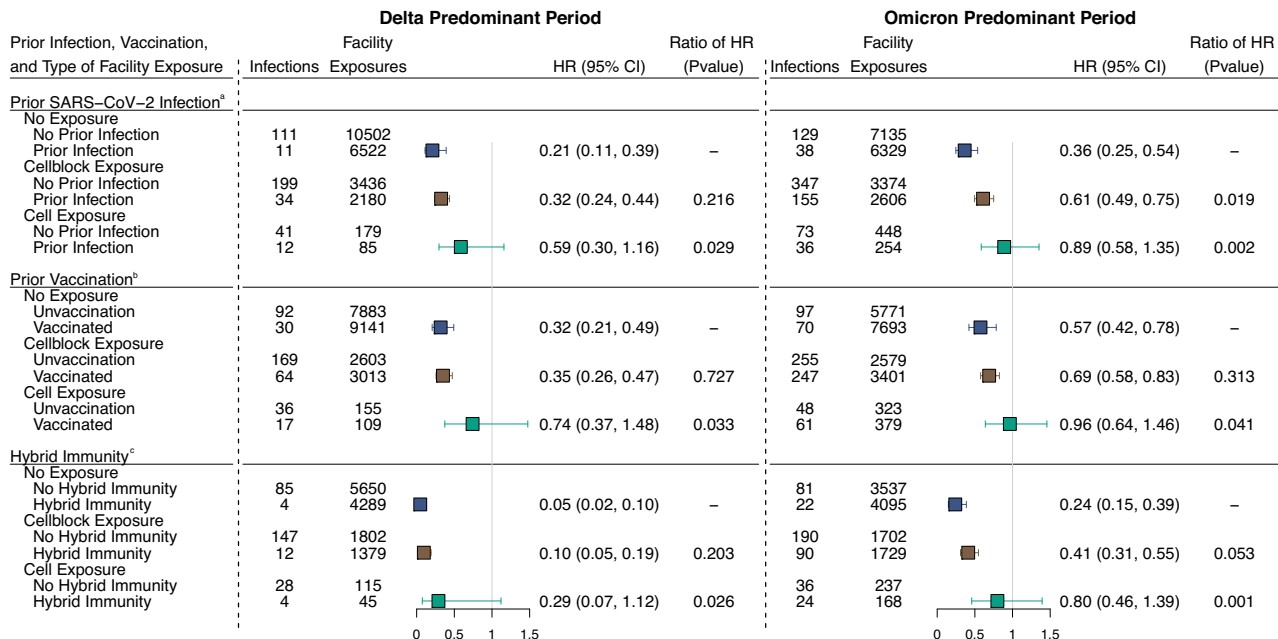

**Fig. 4 | Effectiveness of prior infection vaccination, and hybrid immunity on SARS-CoV-2 infection among residents of Connecticut Department of Correction facilities between June 15, 2021, and May 10, 2022, by documented exposure status.** Forest plot depicting the association between prior infection, vaccination, and hybrid immunity and the risk of SARS-CoV-2 infection by facility exposure type. Residents were classified as having a cell exposure event (green) the day their cellmate tested positive, having a cellblock exposure event (brown) the day a resident of their cellblock but not cell tested positive, and having an event without documented exposure (navy) if no one in their cellblock tested positive. Cell exposure events that occurred within 14 days following a prior cell exposure event were excluded. Cellblock exposure events and events without documented exposure that occurred in the 14 days following a cellblock or cell exposure event were excluded. Associations were examined using Cox Proportional Hazard Models stratified by cellblock with robust standard errors. Each model was adjusted for age, date of exposure, race, room size, and model (**a**) was adjusted for vaccination status and model (**b**) was adjusted for prior infection status. Model (**c**) was limited to residents with hybrid immunity or residents without a record of prior infection or vaccination. Prior infection was defined as a recorded positive SARS-CoV-2 test ≥90 days before the event and vaccination was defined as the receipt of ≥1 dose before the event. Hybrid immunity was defined as a record of both a prior infection and ≥1 vaccine dose. Boxes indicate estimated hazard ratio (HR) point values and whiskers indicate 95% confidence intervals (Delta: n = 17,024 no exposure events, 5616 cellblock exposure events, 264 cell exposure events; Omicron: n = 13,464 no exposure events, 5980 cellblock exposure events, 702 cell exposure events). The ratio of HRs refer to the p-value comparing the HR following cellblock or cell exposure events to the HR following events without documented exposures, estimated using two-sided z-tests. No multiple testing adjustment was performed. Unadjusted results in Supplement Table 7.

without documented exposures and lowest following cell exposure events for each additional sensitivity analysis (Supplement Figs. 8, 12, 15, 17; Unadjusted estimates: Supplement Tables 12, 16, 19, 21).

### SARS-CoV-2 exposure specific effects of prior infection and vaccination on Infectiousness

As a secondary analysis, we hypothesized that the prior infection and vaccination status of the index cases may influence transmission. We examined this by restricting our sample to cellblock and cell exposure events and comparing the hazards of infection when the index case had and did not have the immunizing event of interest. During the Delta period, the prior infection history of the index case was associated with a non-significantly higher hazard of SARS-CoV-2 transmission following cellblock exposure events (HR: 1.96 [0.93–4.12]) and a non-significantly lower hazard following cell exposure events (HR: 0.91 [95 CI: 0.20–4.18]). The vaccination status of the index case was associated with a non-significantly lower hazard of SARS-CoV-2 transmission among cellblock exposure events (HR: 0.75 [0.18–3.12]) and cell exposure events (HR: 0.71 [0.26–1.93]; Fig. 5).

During the Omicron period, the prior infection status of the index case was associated with a non-significantly lower hazard of SARS-CoV-2 transmission following cellblock exposure events (HR: 0.52 [0.27–1.03]) and cell exposure events (HR: 0.72 [95 CI: 0.25–2.03]). The vaccination history of the index case was associated with a non-significantly lower hazard of SARS-CoV-2 transmission among cellblock exposure events (HR: 0.55 [0.24–1.24]) and cell exposure events (HR: 0.52 [0.20–1.36]; Fig. 5).

### Discussion

Leveraging the controlled social structure and detailed epidemiological data of correctional facilities, we found that residents with close (cell) exposures and moderate (cellblock) exposures to SARS-CoV-2 infected residents had a significantly higher risk of becoming infected with SARS-CoV-2 than residents without a documented exposure during Delta and Omicron periods. Further, we found that prior SARS-CoV-2 infection, COVID-19 vaccination, and hybrid immunity (prior infection and vaccination) significantly reduced the risk of infection among residents with cellblock exposures and without documented exposures, but not residents with cell exposures during both periods. Finally, we found that the vaccination status of the index case was associated with a non-significant reduction in the risk of secondary SARS-CoV-2 cases following cell and cellblock exposures during Delta and Omicron periods.

Our findings indicate that exposure to an infected resident in a cell or cellblock significantly increased the risk of becoming infected with SARS-CoV-2 and supports the benefit of contact tracing within the cell and cellblock of infected residents. These findings held irrespective of the period when differing variants were circulating. However, the magnitude of the cell exposure effect was smaller during the Omicron period than the Delta period, potentially due to the increased transmissibility of the Omicron variant[25–27]. Despite the observed decline in

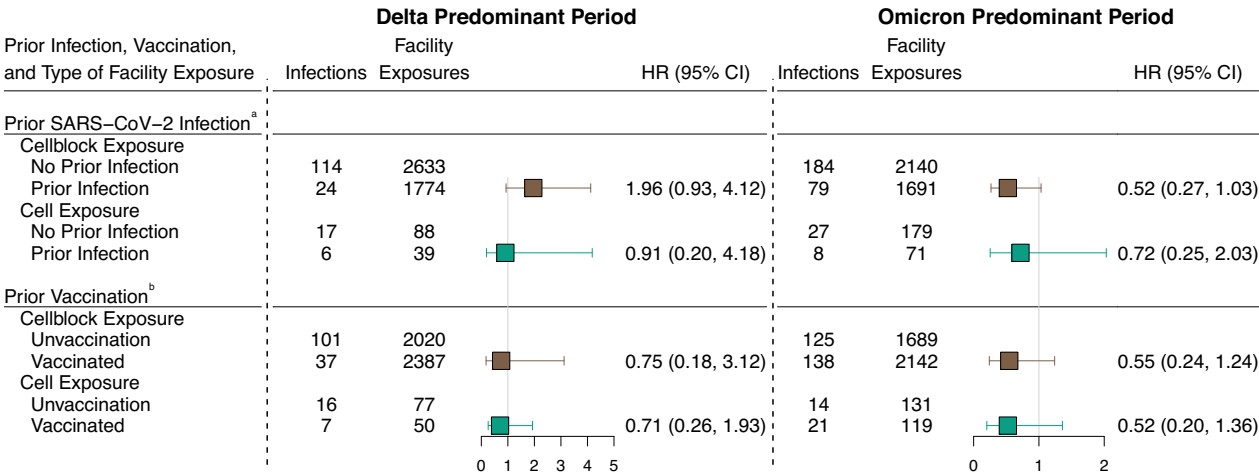

**Fig. 5 | Effectiveness of prior infection and vaccination status of index cases on SARS-CoV-2 transmissibility among of residents of Connecticut Department of Correction facility between June 15, 2021, and May 10, 2022, by documented SARS-CoV-2 exposure status.** Forest plot depicting the association between vaccination and prior infection and the risk of subsequent SARS-CoV-2 infection by documented SARS-CoV-2 exposure status. Residents were classified as having a cell exposure event (green) on the day their cellmate tested positive, having a cellblock exposure event (brown) the day a resident of their cellblock but not cell tested positive, and having an event without documented exposure if no one in their cellblock tested positive on a given day. Cell exposure events that occurred within 14 days following a prior cell exposure event were excluded. Cellblock exposure events and events without documented exposures that occurred in the 14 days following a cellblock or cell exposure event were excluded. Residents were classified as being vaccinated if they had received at least one vaccine dose.

Associations were examined using Cox Proportional Hazard Models stratified by housing cellblock with robust standard errors. The models were adjusted for (a) age, date of exposure, race, room size, vaccination and prior infection status of the susceptible resident, and vaccinated status of the index case (limited to exposed residents), (b) age, date of exposure, race, room size, vaccination and prior infection status of the susceptible resident, and prior infection status of the index case (limited to exposed residents). Prior infections were defined as a recorded positive SARS-CoV-2 test at least 90 days prior to the event and vaccination was defined as the receipt of at least one dose prior to the event. Boxes indicate estimated hazard ratio (HR) point values and whiskers indicate 95% confidence intervals (Delta: $n = 4407$ cellblock exposure events, 127 cell exposure events; Omicron: $n = 3831$ cellblock exposure events, 250 cell exposure events). Unadjusted results in Supplement Table 8.

effect size, cellblock or cell exposures increased the hazard of infection by 3.3 and 4.7 times during the Omicron period, respectively. These findings speak to the continued need for contact tracing within correctional facilities and other high-density settings, including nursing homes, and suggest that contact tracing should not be limited to residents of the same cell but include residents that interact during recreation and meals, as is the case among residents of the same cellblock within Connecticut DOC run facilities.

During both the Delta and Omicron periods, we found that neither prior infection, nor vaccination, nor hybrid immunity provided significant levels of protection against SARS-CoV-2 infection following cell exposure events and that the levels of protection were significantly smaller following cell exposure events than following events without documented exposures. Further, despite having a limited sample during the period of Delta predominance, we observed similar gradients in the level of protection offered by prior infection, vaccination, and hybrid immunity against symptomatic infection. These findings provide empirical evidence that, while accounting for factors thought to be associated with vaccine acceptance and infection, the protection offered by prior infection, vaccination, and hybrid immunity, appears to be leaky. They suggest that there may be an additional mechanism, based on the intensity of the infectious exposure, which may explain observed, partial levels of immunity conferred by infection and vaccination, in addition to factors such as variant-specific immune escape, waning immunity and reduced effectiveness in specific subpopulations, such as older people[28–30].

Beyond providing an evidence base for the mechanism by which prior SARS-CoV-2 infections and COVID-19 vaccines confer immunity, these findings have broad implications on SARS-CoV-2 transmission modeling, vaccine effectiveness analyses, and prevention strategy development. Though most SARS-CoV-2 transmission models likely incorporate a simplifying assumption that the vaccines provide leaky

protection[31,32], the validity of this assumption has not been previously documented with empirical data and model parameterization based examinations were found to be inconclusive[31]. Thus, our findings provide evidence that this assumption may be valid. They also indicate that when estimating future disease burdens under scenarios of defined exposures, modelers may need to account for the reduced effectiveness of prior infections and vaccinations among modeled participants with prolonged, close exposures[31,32]. Furthermore, theoretical studies have demonstrated that the efficacy of leaky vaccines is underestimated by common study designs, which may contributes to variation in observed vaccine effectiveness across settings[17,33].

These findings also suggest the benefit of layered interventions in general, and particularly within densely packed social settings. In the presence of leaky vaccines, non-pharmaceutical interventions have been proposed in tandem with vaccination in order to reduce exposure and mitigate infection spread[34]. Such interventions may include social distancing, quarantine and isolation, masking, and improved ventilation and airflow[35,36]. While our findings are obtained from the investigation of a correctional facility system, in the presence of a leaky vaccine, layered interventions may afford a benefit in other congregate settings and community settings where prolonged, close contact with infected people may occur, such as mass gatherings.

If the protection offered by vaccination is indeed leaky, the increased transmissibility of the Omicron variant may have contributed to the well documented decline in the effectiveness of vaccination during periods of Omicron predominance[26,27,37]. In alignment with prior studies and this speculation, we observed lower levels of protection during the Omicron period. Though this decline has been primarily attributed to variant specific immune escape due to the large number of mutations present in the spike protein[37–39], the high transmissibility of the Omicron variant may have resulted in high enough exposure levels (pathogen pressure) in the community to enhance the

effects of leaky protection in populations that experienced prior SARS-CoV-2 infection and/or vaccination. This speculation invokes the question of whether layered interventions would provide increased benefit in general settings, not just within densely packed settings, when highly transmissible variants are circulating.

We found that hybrid immunity offered the highest level of protection, followed by prior infection. While this finding aligns with prior studies[40–42], the observed differences in the level of protection may reflect the recency of prior infections compared to vaccination (due to an artificial truncation of the time since infection resulting from the absence of community infection data). However, the absence of data on infections that occurred in the community prior to incarceration may have resulted in misclassification and led to conservative estimates for the effectiveness of prior infection. To examine the impact of these missing data, we conducted a sensitivity analysis limited to residents incarcerated since June 15, 2021 (study beginning). Relative to the primary analysis, we did not observe a specific directional shift in the effectiveness estimates, and we observed a similar gradient in the levels of protection by facility exposure. Though, data limitations prevented us from performing this analysis among people incarcerated since the beginning of the pandemic, infection-induced seroprevalence estimates from Connecticut suggest that only a small proportion of the population had been infected as of July 2021 (4.7–19.7%) and the residual bias is likely limited[43,44].

Our effect estimates on infectiousness during the Omicron period are in alignment with a prior study by Tan et al.[45]. This study conducted in California correctional facilities, found that the prior infection and vaccination status of an index case reduced the risk of transmission by 40% (20–55%) and 22% (6–36%), respectively. Similarly, we found that the risk of transmission following a cell exposure was 0.72 (0.25–2.03) times lower among infected residents with a prior infection and 0.52 (0.2–1.36) times lower among infected residents with a history of vaccination than residents without a prior infection or history of vaccination. Though our precision prevents us from making broad conclusions from these findings, they support the findings of Tan et al.[45].

We acknowledge that our study was subject to several limitations. A key potential limitation stems from testing related differences following events with and without documented exposures. Though the Connecticut DOC has, and continues to, conduct intensive COVID-19 testing, testing is more common among residents with an infected cellmate than residents without a documented exposure (Fig. 1C) and may result in an overestimation of the effect of cell or cellblock exposures. While we did observe an attenuation towards the null following the restriction to people who were tested during follow-up, this restriction did not remove the observed gradient in the levels of protection conferred by prior infection, vaccination, or hybrid immunity by facility exposure. In addition to testing frequency, we were concerned that differences in the proportion of tests conducted as a result of symptoms may have introduced bias into our analysis (Supplement Fig. 4). However, very few tests were performed for symptomatic reasons and we continued to find the levels of protection to be highest following events without documented exposures and lowest following cell exposure events.

Another potential source of testing related bias stems from the contact tracing protocol. During the study period, the contact tracing protocol remained consistent and followed the recommendations of the CDC (see Supplement DOC COVID-19 Testing)[46]. The protocol stated that close contacts should be tested 5 days after contact, however, variation in the exact day of testing was probable. Because rapid antigen tests (the test used for contact tracing) are highly sensitive to the viral load, residents tested too soon or long after a contact may have a false negative test, especially if they have a history of prior infection or vaccination (which reduces the viral load)[47]. If variation in the time between contact and testing existed between events without documented exposure, cellblock exposure events, and cell exposure

events, upward or downward bias may have been introduced. Testing prioritization is another potential source of bias from contact tracing. However, due to the testing capacity, contact tracing was performed among symptomatic and asymptomatic residents with or without history of prior infections or vaccination and no prioritization was required.

Our analysis was conducted in a single DOC system and the findings may not be generalizable to all correctional facility settings. Further, we did not have testing or infection data for staff, nor did we have comorbidity and masking data for residents and symptom data for RT-PCR tests. The absence of comorbidity data may result in biased estimates of effectiveness as residents with comorbidities are more likely to become vaccinated and may be more or less likely to become infected with SARS-CoV-2, depending on differential behaviors. However, through adjusting for age and race, we may have accounted for a part of the confounding effect of comorbidities. Due to the absence of symptom data for RT-PCRs, we defined symptomatic infection as a symptomatic rapid antigen test, thus assuming RT-PCR detected infections were asymptomatic. Data limitations prevented us from examining the leakiness of immunity-conferring events against severe outcomes and future analyses with complete symptomatic and severe outcomes data should examine leakiness relative to these outcomes.

Due to sample limitations, we were unable to stratify vaccination and prior infection status by time since vaccination. However, we found no significant difference in the time since prior infection or vaccination between residents with events with and without documented exposures. Behavioral differences between people with prior infections or who have been vaccinated may differ from people without a prior infection or vaccination. While this may have led to either an over or under estimation of the effect of cell or cellblock exposures, it should not have driven our findings suggesting leakiness. Finally, while we conducted numerous sensitivity analyses to examine the robustness of our findings, we were unable to account for all potential sources of bias at the same time and residual bias may be present within our findings.

This study provides empiric evidence that COVID-19 vaccination and prior infection confer exposure dependent ("leaky") protection against SARS-CoV-2 infections. The findings support the use of leaky vaccine parameters in SARS-CoV-2 transmission modeling and indicate the need for modelers to account for the reduced protection conferred by prior infection and vaccinations among people with prolonged, close exposures. Further, our findings suggest the need for layered interventions to mitigate SARS-CoV-2 spread, especially within dense settings, such as congregate settings, and in settings where prolonged contact is likely, such as households with infected people.

## Methods

### Population and data

We conducted a rolling matched cohort analysis among residents of Connecticut DOC facilities who were incarcerated between June 15, 2021, when Delta became the predominant variant in Connecticut according to sequenced clinical samples, and May 10, 2022[24]. Resident demographic (age, race, gender), housing (daily facility, cellblock [block of cells or dorm], cell or dorm, and bunk), and COVID-19 testing, and vaccination data were extracted from DOC maintained databases containing data collected as part of routine SARS-CoV-2 surveillance. Testing records included all rapid antigen (primarily BinaxNOW) and RT-PCR (primarily analyzed by Quest) tests administered within a DOC operated facility since the beginning of the pandemic. We excluded residents who never spent a night in a cell with at least one cellmate, spent <14 days incarcerated, or resided exclusively within a restricted housing cellblock.

The research was performed by researchers at the Connecticut Department of Correction and Yale University (located within the state of Connecticut). All roles and responsibilities were determined by the

collaborating researchers ahead of analysis and the questions raised were done so collectively. The study was determined to be a public health surveillance activity by the Yale University Institutional Review Board and exempt from review (ID: 2000031675). The study results do not stigmatize, incriminate, or discriminate the participants. Our citation list includes research previously published by this collaborative group regarding testing policies within the CT DOC and seroprevalence studies from CT (not from this group).

### Department of correction COVID-19 protocols

Since the introduction of COVID-19 in the winter of 2019–2020, the Connecticut DOC has implemented numerous COVID-19 prevention strategies including testing (rapid antigen and RT-PCR), masking, isolation/quarantine, and vaccination. As part of their COVID-19 mitigation strategy, the DOC restricted the interaction of residents during meal and recreation time to residents of the same cellblock. Thus, during the study period, residents of the same cellblock interacted with other residents of their cellblock during meal and recreation times but, unless their employment required them to move throughout the facility, the residents did not interact with residents of different cellblocks. However, DOC staff continued to move throughout the facilities and were placed in different cellblocks on different days.

Masks are required for all residents while outside of their cell or, if residing in a dorm, moving around their dorm. This is analogous to a non-incarcerated person wearing a mask while socializing in public but not having to wear a mask within their home. Testing with RT-PCRs was and continues to be conducted primarily for mass testing. Testing with rapid antigen test was and continues to be conducted for five primary reasons: intakes/transfers, symptomatic, employment, and contact tracing[48,49]. Among residents of cells, contact tracing included testing all residents of the same cell as the infected resident and residents of the same cellblock or facility but only if close contact was reported by the infected resident. Close contact was defined in accordance with the CDC definition (being within six feet for at least 15 min within a 24-h period)[46]. Resident testing as part of mass screening is considered optional but regular testing is required for many within facility jobs as well as some community facing jobs. The specific testing requirements vary by position. Residents who test positive for SARS-CoV-2 are moved to isolation the day they test positive. For details on testing see Supplement: DOC COVID-19 Testing.

A detailed description of the vaccination program can be found elsewhere[49,50]. Briefly, the DOC began their COVID-19 vaccination program on February 2nd, 2021 and provided vaccines to residents who qualified for vaccination according to state-defined eligibility and were not actively infected. Residents who were partially vaccinated were offered second or subsequent doses of the corresponding vaccine. Vaccinations received prior to incarceration were verified using CT WiZ, Connecticut's COVID-19 vaccine registry[50].

### Sample, type of SARS-CoV-2 exposure, follow-up and matching

For each resident, we identified the days they were housed in a cell. We excluded the first 14 days a person was in the study along with days a resident was housed in a restricted housing cellblock, had an undefined housing location, or did not have at least one roommate. Additionally, to prevent the inclusion of the same infection more than once, we excluded resident days in the 90 days following a positive SARS-CoV-2 infection.

On each included day, residents were classified as having one of three facility structure defined SARS-CoV-2 exposure event types: cell exposure event, cellblock exposure event, or events without documented exposure. We classified a resident as having a cell exposure event if at least one of their cellmates tested positive for SARS-CoV-2, a cellblock exposure event if at least one resident of the same cellblock but different cell tested positive for SARS-CoV-2, or an event without documented exposure if no one in their cellblock tested positive for

SARS-CoV-2 (Supplement Fig. 1). We excluded cell exposure events that occurred within 14 days of a prior cell exposure event and cellblock exposure events and events without documented exposures that occurred within 14 days of a prior cellblock or cell exposure event. Further, to remove the risk of including multiple events without documented exposures from the same resident during a 14-day period, we randomized the sample of residents with events without documented exposures and dropped all days for each person within 14 days of the selected date. Following this exclusion, we cluster matched the cell exposure events, cellblock exposure events and events without documented exposures on facility (exact) and calendar date (+/− 7 days). This ensured that each exposure group was observed at the same time and in the same facility.

Residents were defined as becoming infected with SARS-CoV-2 if they tested positive during the 14 days following inclusion[45]. We censored resident time on the date of release or death or when a resident became exposed at a more proximal level (ex. resident with a cellblock exposure was exposed within their cell). This sampling schematic allowed for residents to be included in the analysis multiple times for the same or different facility exposure statuses. The sample was then stratified by variant predominance within Connecticut (Delta: June 15, 2021, through December 12, 2021; Omicron: December 13, 2021, through the end of the study [May 10, 2022])[24]. We stratified this analysis by variant predominance due to differences in the transmissibility of the variants and the levels of protection offered by prior infections and vaccinations against the variants[15,26,38].

### Prior infection, vaccination and hybrid immunity status

We identified the prior infection, vaccination, and hybrid immunity status of residents with cell exposure events, cellblock exposure events, and events without documented exposure. Additionally, we identified the prior infection, vaccination, and hybrid immunity status of the index cases (infected residents that resulted in all cell and cellblock exposures). We classified a person as being vaccinated if they had received at least one COVID-19 vaccine dose, regardless of the brand or time since the dose was administered. We defined a prior infection as a positive, recorded SARS-CoV-2 rapid antigen or RT-PCR test collected in a DOC facility at least 90 days prior to the date of inclusion. Hybrid immunity was defined as a person having received at least one vaccine dose and having had at least one prior infection as of the date of inclusion.

### Statistical analysis

We visually summarized the vaccine coverage, number of COVID-19 tests and number of SARS-CoV-2 infections recorded among DOC residents during the study period. We summarized the resident characteristics of cell exposure events, cellblock exposure events, and events without documented exposures using medians, first and third quartiles, counts and percentages. Resident gender is evaluated at intake by correctional officers and designated based on genitalia (observed during intake strip search) and governmental documents (passport, driver's license, and birth certificate). We compared the time since last prior infection and vaccine dose using linear models adjusted for age, race, time of inclusion, room size, and cellblock (mirroring the adjustment factors included in the primary analyses, see Facility Exposure Specific Effects of Prior Infection, Vaccination, and Hybrid Immunity on Susceptibility). Data cleaning, management, and analyses were conducted in R version 4.2.1.

### Association between facility exposure and SARS-CoV-2 infection risk

We estimated the association between known SARS-CoV-2 exposure and SARS-CoV-2 infection risk using a facility stratified Cox Proportional Hazards model with an outcome of test positive SARS-CoV-2 infection, a primary exposure of type of facility exposure (cell

exposure events, cellblock exposure events, or events without documented exposure). Further, to account for the correlation of events among people residing within the same cellblock, we estimated confidence intervals using robust standard errors. The model was adjusted for the following a priori selected potential confounders (Supplement Fig. 18): calendar time (continuous), age (continuous), self-identified race (non-Hispanic Black, non-Hispanic White, Other), and room and cellblock size (continuous). Continuous variables were modeled flexibly using natural splines. Though gender is a potential confounder, facilities contain residents of only a single gender and all contrast was eliminated by facility stratification. Significance was defined with an alpha of 0.05 and determined using two-sided z-tests.

### Facility exposure specific effects of prior infection, vaccination, and hybrid immunity on susceptibility

We estimated the association between prior infection and infection risk using a cellblock stratified Cox Proportional Hazards model with robust standard errors, an outcome of SARS-CoV-2 infection, a primary exposure of prior infection history, and an interaction term between facility exposure type and prior infection history. The model was adjusted for the calendar time (continuous), age (continuous), staff assigned race, room size (continuous), and the vaccination history of the susceptible resident. The effect of vaccination was examined using the same model but with an exposure of vaccination instead of prior infection and an adjustment factor of prior infection. We estimated the effect of hybrid immunity using the same model but with an exposure of hybrid immunity. This analysis was restricted to residents with either hybrid immunity or no history of prior infection or vaccination. We tested if the hazard ratios of cell and cellblock exposure events were significantly different than the hazard ratio for events without documented exposures using two-sided z-tests and defined significance with an alpha of 0.05. As a secondary analysis, we estimated the effect of prior infection, vaccination, and hybrid immunity on the hazard of symptomatic infection. Due to an absence of symptomatic data for RT-PCR tests, we defined symptomatic infection as a positive rapid antigen test collected from a symptomatic resident. We used the same models for these analyses as we did for the infection outcome analyses.

### Facility exposure specific effects of prior infection and vaccination on infectiousness

As a secondary analysis, we were interested in evaluating the impact the index cases' history of prior infection and vaccination had on the risk of SARS-CoV-2 infection among residents with cellblock and cell exposure events. For this analysis, we restricted our sample to residents with facility exposures. If residents were exposed to multiple index cases on the same day, we restricted to residents who were exposed to index cases with the same prior infection and vaccination histories. We estimated the effect of prior infection on infectiousness using a cellblock stratified Cox Proportional Hazards model with robust standard errors. The model had an outcome of SARS-CoV-2 infection, an exposure of the index case's prior infection history, and an interaction term between facility exposure type and the index case's prior infection history. The model was adjusted for the same factors as the susceptibility analysis model and the vaccination history of the index case. The effect of vaccination was examined using the same model but with a primary exposure of the index case's vaccination history instead of prior infection history.

### Sensitivity analyses

To test the robustness of our findings to alternative study design, data cleaning, and modeling assumptions, we conducted multiple sensitivity analyses (Supplement: Sensitivity Analyses). Of particular concern was biases due to differences in testing frequency or reasons for testing among residents with and without cell or cellblock exposures.

To examine the impact of potential testing bias, we conducted two sensitivity analyses. First, we restricted to residents tested during follow-up. Second, because symptoms are associated with the level of protection conferred by prior infections and vaccinations, we performed an additional analysis which excluded residents tested due to symptoms (reason for testing listed as symptomatic).

In addition to concerns around testing related biases, we were concerned that the absence of community infection data may have resulted in biased effectiveness estimates for prior infections. We examined the impact of this missing data by limiting our sample to people incarcerated since the beginning of the study (June 15, 2021). Additionally, we were concerned that we may have overestimated the association between documented SARS-CoV-2 exposure and infection risk by including residents' exposure to multiple infected residents in their cell or cellblock on the same day. To examine this, we conducted a sensitivity analysis restricted to cell and cellblock exposure events where only one index case was observed. Further, to ensure that our decision to include residents without recent negative tests did not drive our findings, we conducted a sensitivity analysis restricted to residents who tested negative in the prior 5 days. Additionally, we wanted to ensure our exposures were temporally linked to observed infections. To do so, we conducted two sensitivity analyses: one excluding the first 2 days of follow-up, and one limiting follow-up to 9 days. For a detailed description of sensitivity analyses performed see Supplement: Sensitivity Analyses.

### Reporting summary

Further information on research design is available in the Nature Portfolio Reporting Summary linked to this article.

## Data availability

The data used in this study belongs to the Connecticut Department of Correction and cannot be shared publicly because of the presence of potentially identifiable health and resident information. Qualified researchers may request for de-identified, patient level data by contacting the corresponding author with a detailed description of the research question and setting up a data use agreement with the Connecticut Department of Correction.

## Code availability

Code generated to conduct the statistical analyses is available in the following repository: https://github.com/lindm89/CT_DOC_Dose_Effect_Vax.git[51].

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

## Acknowledgements

We are grateful to the individuals who provided the information for the
analyses and thank the staff of the Connecticut Department of Correc-
tions for their efforts in the response to the COVID-19 pandemic and in
collecting data for the study. We also thank Ryan Borg and Dava Flowers-
Poole for their assistance in the coordination of the study. This work was
supported by a contract from the Connecticut Department of Public
Health (Emerging Infections Program 2021-0071 to A.I.K.), the Raj and
Indra Nooyi Professorship (to A.I.K.), the Sendas Family Fund (to A.I.K.),
the National Institutes of Health (R01 AI174105 to A.I.K. and
1K99AI177945-01 to M.L.L.) and the Merck Investigator Studies Program
(to W.L.S. and A.I.K.). The funders did not have a role in the design or
implementation of the study nor the decision to publish the study. The
study and its findings are the responsibility of the authors and do not
reflect the views of the Connecticut Department of Correction.

## Author contributions

M.L.L. and A.I.K. have full access to all the data in the study and take
responsibility for the integrity of the data and the accuracy of the data
analysis. M.L.L., A.I.K., R.P.R., and B.S.K. conceived the study. The data
was collected and processed by M.L.L., M.D., A.J.H., M.L., B.S.K., and
R.P.R. M.L.L., M.D., M.D.T.H., D.A.T.C., R.T., and S.L. performed the
analysis. M.L.L., A.I.K., B.S.K., M.D.T.H., D.A.T.C., J.R.A., I.Y., S.B.O.,
W.L.S., and R.R. drafted the manuscript. All authors provided critical
review of the results and contributed to manuscript revision. A.J.H.,
M.L., B.S.K., R.P.R., and A.I.K. supplied administrative, technical, and
material support. Supervision was provided by B.S.K., R.P.R., D.A.T.C.,
and A.I.K.

## Competing interests

A.I.K. is as an expert panel member for Reckitt Global Hygiene Institute,
and a consultant for Tata Medical and Diagnostics and Regeneron
Pharmaceuticals and has received grants related to COVID-19 research
outside the scope of the proposed work from Regeneron Pharmaceu-
ticals and Tata Medical and Diagnostics. W.L.S. was an investigator for a
research agreement, through Yale University, from the Shenzhen Center
for Health Information for work to advance intelligent disease prevention
and health promotion; collaborates with the National Center for Cardi-
ovascular Diseases in Beijing; is a technical consultant to Hugo Health, a
personal health information platform, and co-founder of Refactor
Health, an AI-augmented data management platform for healthcare; and
has received grants related to COVID-19 research outside the scope of
the proposed work from Regeneron Pharmaceutical. The other authors
declare no competing interests.

## Additional information

**Supplementary information** The online version contains
supplementary material available at

Margaret L. Lind or Albert I. Ko.

**Peer review information** *Nature Communications* thanks Eyal Leshem,
Alicia Kraay and the other, anonymous, reviewer for their contribution to
the peer review of this work. A peer review file is available.

