## [Peer Review File · Nature Communications]

Evidence of Leaky Protection Following COVID-19 Vaccination and SARS-CoV-2 Infection in an Incarcerated PopulationREVIEWER COMMENTS

Reviewer #1 (Remarks to the Author):

RE: NCOMMS-23-09666-T

An interesting investigation of vaccine effectiveness on different exposures – cell / cell block / no identifiable exposure conducted to specifically assess protection leakiness. The method design is appropriate, and there are few opportunities which would provide similar "natural experiment" settings for such assessments. The authors conducted sensitivity analyses to assess the impact of inherent limitations and biases.

My main interest is why the authors did not present an analysis of vaccine effectiveness against symptomatic disease. With most testing done to asymptomatic persons, as part of contact tracing and mass screening (figure 1d), the meaning of a positive test in this setting is more challenging to interpret for public health action.

I have few specific comments:

L100 – "Delta variant (June 15 to December 13, 2021) and Omicron variant (December 13, 2021 to May 10, 2022)". The authors need to specify mutually exclusive periods (can't have December 13 on both periods).

L364 – See above – may explain why VE against symptomatic infection was not presented but should be specified here.

L388 – Severe outcomes normally refers to severe disease, hospitalization or death. Symptomatic disease is not considered severe outcome.

Reviewer #2 (Remarks to the Author):

The manuscript "Evidence of Leaky Protection Following COVID-19 Vaccination and SARS-CoV-2 Infection in a US Correctional Facility Population" by Margaret Lind et al. summarizes an analysis of surveillance data to explore SARS-CoV-2 transmission and its covariates among a vulnerable population in a congregate setting. While the objective of putting surveillance data to work to protect the health of at-risk populations is commendable, the results presented are neither adequately justified by the analysis, nor supportable by the data. Further details follow.

The authors acknowledge a number of plausible confounders in the ascertainment of infection, related to the relatively low testing rate and to biases in testing. Several of the confounders are analytically addressed, one at a time, through "sensitivity analyses"; the results hint that a fully and appropriately adjusted analysis may be either underpowered or otherwise unable to support the principal claim presented. In any case, it remains unclear to this reviewer if the data (collected for public health surveillance and response, not for an analytical study) can be adequately adjusted for the multitude of biases, and incompleteness in some key variables (prior infection, and time since vaccination, for instance) for the intended analysis.

The large discrepancy between the proportion with prior infection in the study population compared to that in the community (as reported in serosurveys), especially during the Omicron wave, gives one pause; is there a plausible explanation here other than incomplete ascertainment?

A relatively high proportion of infections were detected through contact tracing. Detection of secondary cases through contact tracing and follow-up testing is very sensitive to the details of the protocol -- the definition of contacts, prioritization for testing, the pace of the operation, etc. Each of those details, none described, may confound the analysis.

The line of analysis presented in the manuscript is poorly supported by the data, but may not do justice to the potentially impactful public health story the data hold. A more descriptive analysis, however, may be quite helpful in understanding the effectiveness of different components of a contact tracing protocol in high-risk congregate settings, the dependence of incidence within a correctional facility on incidence in the surrounding community, etc. I would strongly recommend reshaping and redirecting the manuscript to a more public health practice oriented audience (through MMWR, or a similar journal), where the results stand to have substantial impact on public health.

Reviewer #3 (Remarks to the Author):

In this manuscript, the authors make use of a prison population in Connecticut to estimate the extent of protection conferred by vaccination or prior infection against subsequent infection by type of exposure, including cell sharing, block exposure, and unknown exposure. The authors show that protection appeared to be dose-dependent, with protection being more pronounced against more weaker exposures. These general results also held in various sensitivity analyses. The methodology for the analysis is sound. In general, this is a compelling paper that provides clear evidence of dose-dependent protection for COVID-19 and is unique and exciting for the field.

I have several suggestions for the authors to consider:

1. The authors have convinced me that the general pattern of results is constant regardless of how they subset their sample (for example, results are similar when restricted to inmates who tested) but I am not sure why the authors choose to present numerical results on the broadest results in the main text, as these numbers are likely to be more widely used in follow up analyses. Can the authors clarify? Perhaps it would be helpful to at least state the point estimates for the hazard ratios from some of the sensitivity analyses in the main text (maybe the one for inmates who tested).
2. Have the authors investigated whether the joint exposure of prior infection and prior vaccination influences protection? If the authors have sufficient power to look at this question, it would be very useful to present some data on this point as hybrid immunity is becoming increasingly relevant as COVID-19 continues to spread. It seems that there might be sufficient events to make this comparison, particularly during the omicron wave.

3. It might be useful to connect the recency of exposure to the comparative results of prior infection and vaccination. In the text, it looks like the protection conferred by prior exposure is similar to or stronger than that conferred by vaccination, but prior exposure is artificially truncated at time since incarceration whereas vaccine history is not. Thus, the stronger association may be more related to recency of infection.

Minor comment:

1. Please rephrase sentence 1 to say “exposure-dependent (“leaky”) protection against infection.” While this sentence as originally written is accurate for infection, leakiness in general is also defined as it relates to blocking some aspects of infection but not others (ex: blocking severe disease but not infection).

Reviewer Responses

We appreciate the reviewers' thoughtful and insightful reviews. They have meaningfully improved our manuscript. Please find our responses to the editor and reviewer comments below (see our responses in blue). We have also provided an updated manuscript, supplement, tables, and figures with tracked changes as requested. Please note, all line numbers correspond to the clean (non-track changed) word version (not pdf) of the updated manuscript.

REVIEWER COMMENTS

We appreciate the constructive and helpful comments of the three reviewers and have incorporated their recommendations into the manuscript per our point-by-point response.

Reviewer #1 (Remarks to the Author):

RE: NCOMMS-23-09666-T

An interesting investigation of vaccine effectiveness on different exposures – cell / cell block / no identifiable exposure conducted to specifically assess protection leakiness. The method design is appropriate, and there are few opportunities which would provide similar "natural experiment" settings for such assessments. The authors conducted sensitivity analyses to assess the impact of inherent limitations and biases.

Response: Thank you very much for this thoughtful review of our article. We appreciate your comments and suggestions.

My main interest is why the authors did not present an analysis of vaccine effectiveness against symptomatic disease. With most testing done to asymptomatic persons, as part of contact tracing and mass screening (figure 1d), the meaning of a positive test in this setting is more challenging to interpret for public health action.

Response: We agree with the reviewer and have added a secondary analysis evaluating the effects of prior infection and vaccination against symptomatic infection (Methods: Lines: 576-580). We found that, despite limited precision around our Delta estimates, the level of protection was highest following events without documented exposures and lowest following cellblock exposure events (Results: Lines: 208-213, 242-247). Further, we have clarified that the testing listed as symptomatic in Figure 1 represents testing in the presence of recorded symptoms, regardless of reason.

I have few specific comments:

L100 – "Delta variant (June 15 to December 13, 2021) and Omicron variant (December 13, 2021 to May 10, 100 2022)". The authors need to specify exclusive periods (can't have December 13 on both periods).

Response: Thank you for pointing out this oversight. The period of delta predominance was defined as June 15 through December 12th and the Omicron period began on December 13th. We corrected this in the manuscript (Lines: 56-57).

L364 – See above – may explain why VE against symptomatic infection was not presented but should be specified here.

Reviewer Responses

Response: We added a secondary analysis with an outcome of symptomatic infection (See comment 1). Along with the results, we have also added language discussing the limitations of this analysis in the Discussion (Lines: 415-419).

L388 – Severe outcomes normally refers to severe disease, hospitalization or death. Symptomatic disease is not considered severe outcome.

Response: Thank you for bringing this to our attention. We revised the language in the Discussion (Lines: 417-419) to specifically discuss symptomatic infection and severe outcomes separately.

Reviewer #2 (Remarks to the Author):

The manuscript "Evidence of Leaky Protection Following COVID-19 Vaccination and SARS-CoV-2 Infection in a US Correctional Facility Population" by Margaret Lind et al. summarizes an analysis of surveillance data to explore SARS-CoV-2 transmission and its covariates among a vulnerable population in a congregate setting. While the objective of putting surveillance data to work to protect the health of at-risk populations is commendable, the results presented are neither adequately justified by the analysis, nor supportable by the data. Further details follow.

We appreciate Reviewer 2's comments and the many helpful suggestions provided to improve the analyses and the manuscript.

The authors acknowledge a number of plausible confounders in the ascertainment of infection, related to the relatively low testing rate and to biases in testing. Several of the confounders are analytically addressed, one at a time, through "sensitivity analyses"; the results hint that a fully and appropriately adjusted analysis may be either underpowered or otherwise unable to support the principal claim presented.

Response: We agree that there are multiple potential sources of bias in our analysis, and we are unable to address all the sources of bias at the same time due to sample limitations. We have added language to the discussion clarifying that we are unable to account for the different potential sources of bias in combination (Lines: 427-429).

In any case, it remains unclear to this reviewer if the data (collected for public health surveillance and response, not for an analytical study) can be adequately adjusted for the multitude of biases, and incompleteness in some key variables (prior infection, and time since vaccination, for instance) for the intended analysis.

Response: We agree with the concerns raised by the reviewer. To examine the impact of missed prior infections (a concern also raised by reviewer three), we conducted a sensitivity analysis limited to people incarcerated since June 15, 2021 (the beginning of our study). As with the primary analysis, we found the prior infection conferred the highest level of protection following events without documented exposure and the lowest level of protection following cell exposure events (Lines: 227-232, 253-258; Supplemental Figure 10). Due to data limitations, we unfortunately cannot perform this analysis restricted to people incarcerated since the beginning of the pandemic and we cannot exclude that prior infections from this time may not introduce bias. However, previously published seroprevalence estimates from Connecticut suggest that only a small proportion of the population had been infected prior to July 2021 (between 4.7 and 19.7%).¹ We have added a paragraph to the discussion covering this concern (Lines: 356-369).

Reviewer Responses

1. <https://covid19serohub.nih.gov/studies/S-1f88deea-873f-4b2a-8ec0-33041516e86c>

Further, we examined if time since last immunity-conferring event differed among residents with events without documented exposure, cellblock exposure events, and cell exposure events using linear models adjusted for the factors in the effectiveness analysis (age, race, room-size, cellblock, and inclusion date). We did not find significant differences in the time since last prior infection or vaccination between residents with events with and without documented exposures. We describe these findings in the Results section (Lines: 122-125, 136-138) and the limitations section of the Discussion (Lines: 421-423).

The large discrepancy between the proportion with prior infection in the study population compared to that in the community (as reported in serosurveys), especially during the Omicron wave, gives one pause; is there a plausible explanation here other than incomplete ascertainment?

Response: We appreciate you bringing this concern to our attention and acknowledge that compared with CDC reported estimates of seroprevalence for people aged 18-49 in February 2022 (63.7%; Kristies et al.), our proportion of events among residents with prior infections may appear low for the Omicron predominate period. However, our prior infection proportions are not directly comparable to the CDC seroprevalence estimates. In our study sample, we excluded resident-time during the 90 days following a prior infection. Because most infections that drove seropositivity during the Omicron wave occurred in January and February 2022 (see Figure 1), people infected during the peak of the Omicron wave would only reappear in our sample for a short period of time at the end of the study, if at all (February 10, 2022, is 90 days prior to the end of the study on May 10, 2022). Among our study sample, 65.1% of facility events occurred among residents with at least one recorded prior infection by the study end date of May 10, 2022, which is higher than the seroprevalence reported by Kristies et al. (63.7%).

Additionally, the September 2021 seroprevalence estimates from Kristies et al. was around 34% and our proportion of events among residents with prior infections during the Delta period (June 15 – December 12, 2021) ranged between 32% and 38%.

A relatively high proportion of infections were detected through contact tracing. Detection of secondary cases through contact tracing and follow-up testing is very sensitive to the details of the protocol -- the definition of contacts, prioritization for testing, the pace of the operation, etc. Each of those details, none described, may confound the analysis.

Response: The reviewer brings up a good point. We clarified on the DOC contact tracing protocol in the Results (Lines: 65-68) and the Supplement (Supplement: DOC COVID-19 Testing), included the number of tests performed as a result of contact tracing (Lines: 70-72), and provided additional discussion about the potential biases introduced by the contract tracing protocol in the Discussion (Lines: 380-392).

The line of analysis presented in the manuscript is poorly supported by the data but may not do justice to the potentially impactful public health story the data hold. A more descriptive analysis, however, may be quite helpful in understanding the effectiveness of different components of a contact tracing protocol in high-risk congregate settings, the dependence of incidence within a correctional facility on incidence in the surrounding community, etc. I would strongly recommend reshaping and redirecting the manuscript to a more public health practice-oriented audience (through MMWR, or a similar journal), where the results stand to have substantial impact on public health.

Reviewer Responses

Response: We very much appreciate the reviewer's comment on the importance of a descriptive analysis of the testing and contact tracing protocol. We are following the recommendations by analyzing and preparing manuscripts on the effectiveness of different testing protocols and the relationship between within facility and community infection burdens.

Reviewer #3 (Remarks to the Author):

In this manuscript, the authors make use of a prison population in Connecticut to estimate the extent of protection conferred by vaccination or prior infection against subsequent infection by type of exposure, including cell sharing, block exposure, and unknown exposure. The authors show that protection appeared to be dose-dependent, with protection being more pronounced against more weaker exposures. These general results also held in various sensitivity analyses. The methodology for the analysis is sound. In general, this is a compelling paper that provides clear evidence of dose-dependent protection for COVID-19 and is unique and exciting for the field.

Response: Thank you very much for your thorough review of our analysis and manuscript.

The authors have convinced me that the general pattern of results is constant regardless of how they subset their sample (for example, results are similar when restricted to inmates who tested) but I am not sure why the authors choose to present numerical results on the broadest results in the main text, as these numbers are likely to be more widely used in follow up analyses. Can the authors clarify? Perhaps it would be helpful to at least state the point estimates for the hazard ratios from some of the sensitivity analyses in the main text (maybe the one for inmates who tested).

Response: The reviewer raises a very good point. Along with the numerical results for the broadest analysis, we have expanded our results section to include the numerical results for the key sensitivity analyses (Lines: 156-159, 166-168, 187-190, 223-232, 249-258) including the sensitivity analysis limited to tested residents (Lines: 156-159, 187-190, 223-227, 249-253).

Have the authors investigated whether the joint exposure of prior infection and prior vaccination influences protection? If the authors have sufficient power to look at this question, it would be very useful to present some data on this point as hybrid immunity is becoming increasingly relevant as COVID-19 continues to spread. It seems that there might be sufficient events to make this comparison, particularly during the omicron wave.

Response: We agree and have added joint (hybrid) immunity (record of prior infection and vaccination) to the analysis (Lines: 203-205, 238-240; Figure 4). As with prior infection and vaccination, we found the level of protection to be highest following events without documented exposure and lowest for cell exposure events.

It might be useful to connect the recency of exposure to the comparative results of prior infection and vaccination. In the text, it looks like the protection conferred by prior exposure is similar to or stronger than that conferred by vaccination, but prior exposure is artificially truncated at time since incarceration whereas vaccine history is not. Thus, the stronger association may be more related to recency of infection.

Response: This is a very good point, also raised by reviewer 2. To address this concern, we conducted a sensitivity analysis restricted to people incarcerated since June 15, 2021 (the beginning of our study). As with the primary analysis, we found that the effectiveness of prior

Reviewer Responses

infection was higher than vaccination during the Delta period but not following cell exposure events during the Omicron period (Supplement Figure 10). Regardless, we found the effectiveness was highest following events without documented exposures and lowest following cell exposure events. Unfortunately, due to data limitations, we cannot restrict this analysis to people incarcerated since the beginning of the pandemic and cannot fully eliminate this concern. However, previously published data from Connecticut suggests that only about 4.7-19.7% of the population were seropositive as of July 2021. Along with adding this analysis to the results (Lines: 227-232, 253-258; Supplemental Figure 10), we have added a paragraph to the Discussion (Lines: 356-369) covering the potential biases associated with the absence of community infection data.

Minor comment:

Please rephrase sentence 1 to say “exposure-dependent (“leaky”) protection against infection.” While this sentence as originally written is accurate for infection, leakiness in general is also defined as it relates to blocking some aspects of infection but not others (ex: blocking severe disease but not infection).

Response: Thank you for pointing this out. We have updated the sentence as recommended (Line: 19). Additionally, we have added “against infection” to the abstract (line 4).

REVIEWERS' COMMENTS

Reviewer #1 (Remarks to the Author):

The manuscript was revised in line with my comments and suggestions.

Reviewer #2 (Remarks to the Author):

I would like to thank the authors for addressing my concerns on the original manuscript through additional discussion of the substantial limitations in the analysis. The results presented, however, do not reflect the uncertainties arising from the limitations not adjusted for. The extent to which a vaccine is leaky is not just of interest to disease modelers, but also to policymakers. I would strongly suggest noting clearly in the abstract (and title?) that the evidence for leakiness is weak and could not be adjusted for substantial limitations of the dataset, in language unambiguous to a non-modeler.

Reviewer #3 (Remarks to the Author):

I am satisfied with the author responses to my comments and think the paper has been substantially improved in this revision. I particularly appreciate the inclusion of the hybrid immunity analysis.

While I agree with reviewer 2 that the point estimates may be somewhat biased by inherent issues in testing that the authors cannot account for simultaneously, the magnitude of this bias would have to be substantial to erode the findings presented here. Given that the point estimates remain strong and are nowhere near the null for all sensitivity analyses and that the gradient is clearly preserved across all sensitivity analyses, I think the main findings are robust and clearly show a dose-response impact of exposure, allowing the authors to illustrate leaky protection (the primary goal of the paper). The authors have appropriately described these caveats.

Reviewer Responses (second round)

We appreciate the reviewers' positive responses to our previous revisions and have addressed their additional concerns in the updated manuscript. Please see our point-by-point response below in Blue.

REVIEWER COMMENTS

Reviewer #1 (Remarks to the Author):

The manuscript was revised in line with my comments and suggestions.

Response: Thank you.

Reviewer #2 (Remarks to the Author):

I would like to thank the authors for addressing my concerns on the original manuscript through additional discussion of the substantial limitations in the analysis. The results presented, however, do not reflect the uncertainties arising from the limitations not adjusted for. The extent to which a vaccine is leaky is not just of interest to disease modelers, but also to policymakers. I would strongly suggest noting clearly in the abstract (and title?) that the evidence for leakiness is weak and could not be adjusted for substantial limitations of the dataset, in language unambiguous to a non-modeler.

Response: We understand the concern raised by the reviewer and have added language to the abstract to make it clear that we were unable to address all potential sources of bias.

Abstract: Page 2: Lines 14-16 (New text underlines)

“*Although associations may not have been thoroughly adjusted due to dataset limitations, the findings suggest that prior infection and vaccination may be leaky, highlighting the potential benefits of pairing vaccination with non-pharmaceutical interventions in crowded settings.*”

Reviewer #3 (Remarks to the Author):

I am satisfied with the author responses to my comments and think the paper has been substantially improved in this revision. I particularly appreciate the inclusion of the hybrid immunity analysis.

While I agree with reviewer 2 that the point estimates may be somewhat biased by inherent issues in testing that the authors cannot account for simultaneously, the magnitude of this bias would have to be substantial to erode the findings presented here. Given that the point estimates remain strong and are nowhere near the null for all sensitivity analyses and that the gradient is clearly preserved across all sensitivity analyses, I think the main findings are robust and clearly show a dose-response impact of exposure, allowing the authors to illustrate leaky protection (the primary goal of the paper). The authors have appropriately described these caveats.

Response: Thank you very much for your review of the manuscript and support of our analysis.